# The Ubiquitin Switch in Plant Stress Response

**DOI:** 10.3390/plants10020246

**Published:** 2021-01-27

**Authors:** Paymon Doroodian, Zhihua Hua

**Affiliations:** 1Department of Environment and Plant Biology, Ohio University, Athens, OH 45701, USA; pd214110@ohio.edu; 2Molecular and Cellular Biology Program, Ohio University, Athens, OH 45701, USA

**Keywords:** ubiquitin, E3 ligase, protein degradation, stress, drought, thermotolerance, salinity, ROS, innate immunity

## Abstract

Ubiquitin is a 76 amino acid polypeptide common to all eukaryotic organisms. It functions as a post-translationally modifying mark covalently linked to a large cohort of yet poorly defined protein substrates. The resulting ubiquitylated proteins can rapidly change their activities, cellular localization, or turnover through the 26S proteasome if they are no longer needed or are abnormal. Such a selective modification is essential to many signal transduction pathways particularly in those related to stress responses by rapidly enhancing or quenching output. Hence, this modification system, the so-called ubiquitin-26S proteasome system (UPS), has caught the attention in the plant research community over the last two decades for its roles in plant abiotic and biotic stress responses. Through direct or indirect mediation of plant hormones, the UPS selectively degrades key components in stress signaling to either negatively or positively regulate plant response to a given stimulus. As a result, a tightly regulated signaling network has become of much interest over the years. The ever-increasing changes of the global climate require both the development of new crops to cope with rapid changing environment and new knowledge to survey the dynamics of ecosystem. This review examines how the ubiquitin can switch and tune plant stress response and poses potential avenues to further explore this system.

## 1. Introduction

As global climate changes and the occurrence of more extreme weather patterns becomes more frequent, organisms rely much more than ever on effective mechanisms to buffer the extremes of external environmental fluctuations. This is critical to keep a relatively stable internal environment and maintain active metabolic reactions. To survive from a sudden influx of water resulting from devastating hurricanes, for example, requires the sensing, activation, and expression of water-specific genes such as those encoding aquaporin or vacuolar invertase in plants [1,2]. On the other hand, some proteins need to be rapidly removed for proper signaling transduction in a timely response to environmental perturbations. Such responses are extremely crucial to plants due to their sessile lifestyle. One mechanism to deal with protein removal is manipulated by the ubiquitin (Ub)-26S proteasome system (UPS), which is well appreciated for its breadth and depth in regulating the turnover of intracellular proteins [3,4,5,6,7]. Through the UPS, a 76 amino acid polypeptide, known as Ub, is covalently linked to a substrate’s lysine residues [8]. In many cases, a ubiquitylated protein is recognized by the 26S proteasome, a large, ≈2.5 MDa, multi-catalytic ATP-dependent protease complex, for degradation [9]. The molecular mechanism that is required for proper response to both abiotic and biotic stresses had largely remained elusive until the last two decades. Multiple studies have illuminated a network of crosstalk that takes place between the UPS and phytohormones to activate the signaling pathways necessary for plant stress response. In this review, recent discoveries on these roles are discussed, as well as the genome evolution perspective of plant UPS and future directions.

## 2. Protein Turnover by the Plant UPS

Ub is extremely conserved across the kingdoms of eukaryotic organisms. For example, only three amino acids differ between human and plant Ubs [10]. During the process of protein ubiquitylation, the Ub is sequentially activated by E1 Ub-activating enzyme, conjugated onto an E2 Ub-conjugating enzyme, then catalyzed by a specific E3 Ub ligase for being covalently linked to the ε-amino group of a substrate lysine residue [3,4,5,11,12]. Ub itself has seven lysine residues that allow it to attach multiples of itself in a process called polyubiquitylation [8]. While monoubiquitylation can be a traffic signal for the substrate to go throughout the cell or transported into vacuoles or lysosomes for degradation [13,14], the outcome of protein polyubiquitylation is dependent on the topology of poly-Ub chains. For example, if the Ub moieties are connected via their 63rd lysine residues, a process called lysine-63 polyubiquitylation, the target protein often changes activity or directs to new cellular localization [8,15]. However, if the poly-Ub chain is made through the linkage of the 48th lysine residues, known as lysine-48 polyubiquitylation, the resulting poly-Ub chain serves as a degradation signal leading the targeted substrate for turnover in the 26S proteasome [8,16]. 

Although several reviews have highlighted the roles of non-proteolytic ubiquitin signaling in plant stress responses [15,17], Ub signals are more prominent in proteolytic degradation of target proteins by the 26S proteasome. For example, we discovered that the proteasome inhibitor, MG132, can increase the pool of ubiquitylated proteins in *Arabidopsis thaliana* (Arabidopsis hereafter) seedlings up to fourfold, suggesting that greater than 80% of the total ubiquitylated proteins are degraded by the 26S proteasome (unpublished data). The 26S proteasome is a “cylinder-shaped” enzymatic complex composed of one or two 19S regulatory particles (RP) that cap the two ends of a 20S central core particle (CP) [11]. The biochemical role of the 26S proteasome is to degrade ubiquitylated proteins [4,9,11,18,19]. Once a protein is modified primarily through lysine-48 type polyubiquitylation, the poly-Ub chain(s) is bound to shuttle Ub-binding proteins such as the RAD23 family [20], which brings the ubiquitylation substrates to the proteasome. Alternatively, in some cases, the ubiquitylated proteins are directly associated with Regulatory Particle Non-ATPase (RPN) 1, RPN10, or RPN13 of the 26S proteasome in cytoplasm or in the nucleus [9,19]. The bound ubiquitylated proteins are de-ubiquitylated, unwound, and chewed into the multicatlytic CP protease compartment through the RP subunits for being cleaved into short peptides [11], which are ultimately released out of the proteasome in a yet-unknown mechanism. The short peptides are further degraded into free amino acids by numerous peptidases in cells. Although lysine-48 poly-Ub chain(s) serves as the primary degradation signal for ubiquitylation substrates, proteomic studies also suggest that lysine-11 poly-ubiquitylated proteins can be targeted into the 26S proteasome for degradation [21]. Recent structural and enzymatic kinetic analyses even revealed the degradation of substrates modified by lysine-63 poly-Ub chains in the yeast 26S proteasome through direct recognition by RPN1/10 [19]. It remains to be tested whether the plant 26S proteasome can also target this type of substrate.

The selective degradation of different types of poly-ubiquitylated proteins has been recognized as a signal to determine the fate of a ubiquitylated protein [8]. However, the primary contribution to the specificity of the UPS is ascribed to the large group of E3 ubiquitin ligases encoded in plant genomes. Through genome comparison, we have predicted that >600 E3 ubiquitin ligase genes are present in the basal flowering plant *Amborella trichopoda*, and that this group of genes is dramatically expanded in both monocotyle and eudicotyle plants (Table 1) [22]. If all these genes were to be active, it would be expected that over 6% of the entire proteome encoded in Arabidopsis is dedicated to the UPS regulatory pathway [5]. In comparison, the E3 ligase group in other eukaryotic organisms is much smaller except in nematodes. Using the size of *F-box* gene family as an example, while Arabidopsis genome encodes ≈800 members [23], 20 in the budding yeast *Saccharomyces cerevisiae*, 17 in the fission yeast *Schizosaccharomyces pombe*, 27 in the fly *Drosophila melanogaster*, 69 in human, and ≈500 in nematode *Caenorhabditis elegans* have been reported [24,25,26].

On the basis of structural compositions and the conjugation process of the activated Ub moieties, we can divide plant E3 ligases into three major types: plant U-box (PUB), homologous to the E6-AP carboxyl terminus (HECT), and really interesting new gene (RING)-containing E3 ligases [27,28,29,30]. HECT E3s, unlike PUB proteins and RINGs, are the only group of E3 enzymes that are ubiquitylated by the E2 Ub-conjugating enzyme before they recognize and catalyze the ubiquitylation of the substrates. In Arabidopsis, seven HECT enzymes have been identified [31]. PUB proteins, similar to HECT enzymes, make up a small E3 ligase family in non-plant eukaryotic organisms, such as 2 in human and 21 in yeast [32,33]. However, they are significantly expanded in plants [27,34]. For example, genome-wide analysis in tomatoes identified 62 E3 ligases containing a U-box domain [35]. 

RING E3 ligases represent the most complicated and yet mysterious group of proteins in plant genomes. It is composed of both mono-subunit and multi-subunit E3 enzymes. A mono-subunit RING E3 ligase contains a RING motif for interacting with E2 Ub-conjugating enzyme and a substrate recognition region to recruit its substrate. The multi-subunit RING E3 enzyme can be divided into a large group of cullin (Cul)-RING (CRL) E3 ligases [3,36] and a small group of anaphase-promoting complexes (APCs) [37,38]. The plant CRL E3 ligases can be further divided into three major subgroups primarily distinct by their substrate receptors: (1) Skp1-Cul1-F-box (SCF) complexes in which the large group of F-box proteins function as the substrate receptors, (2) Broad complex-Tramtrack-Bric a brac (BTB)-Cul3a/b complexes that use their BTB proteins for recognizing substrates, and (3) DDB1-binding/WD40-Cul4 complexes that direct protein for ubiquitylation via the DDB1-binding/WD40 proteins [3]. Our recent studies suggest that >1000 RING E3 ligase genes are encoded in both monocotyl and eudicotyle plant genomes (Table 1) [22]. Such large expansion of the E3 ligase group in plant genomes provides essential genomic resources for plants combatting various environmental challenges, e.g., pathogen response [17,39,40,41], drought susceptibility [42,43,44,45,46,47], nutrient homeostasis [48,49], and heat tolerance [50,51,52].

## 3. Crosstalk of Ub with Its Two Siblings, ATG8 and SUMO, in Protein Turnover

Along with the UPS, plants also developed alternative degradation systems that involve two Ub-like modifiers, autophagy-related protein 8 (ATG8) and small ubiquitin like-modifier (SUMO). Genomic comparison in 50 plant genomes revealed contrasting duplication processes of these three small proteins [53]. Intronless *Ub* moieties in a *poly-Ub* gene allowed us to postulate a retrotranspostion model of the *poly-Ub* amplification in plant genomes. However, on the basis of syntenic and genome structure organizations, we suggested that *ATG8* and *SUMO* were duplicated in plant genomes through whole-genome duplications and tandem duplications, respectively [53]. Biochemical analysis of Ub, SUMO, and ATG8-medaited protein modifications revealed a similar three-step cascade reaction, involving E1, E2, and E3 enzymes (Figure 1) [54]. Therefore, convergent evolution is present among these systems, likely resulting from the selection pressure of protein turnover that they commonly shared.

The ATG8-mediated autophagy pathway degrades intracellular components in selective or nonselective manners [55,56]. Unlike Ub proteins, ATG8 is targeted to be conjugated with phospholipids, which form a double-membrane vesicle known as autophagosome. The significantly larger volume of autophagosome in comparison with the narrow channel of the proteolytic center in the 26S proteasome differs its roles as a system for bulk and selective recycling of cellular components. In many cases, the autophagosome engulfs large protein complexes/aggregates or damaged organelles and delivers them to the vacuole for recycling. Since organelles are sensitive to stress response, the role of autophagy in stress regulation and organelle recycling has been reviewed elsewhere [55,56,57]. Interestingly, in yeast, selective autophagy has been recognized to target ubiquitylated vacuole membrane proteins for degradation [58,59]. This raised an open question as to whether intracellular membrane proteins are regulated in a similar way in plant cells. Nevertheless, transmembrane proteins on plasma membrane of plant cells have been found to be removed through vacuolar degradation after ubiquitylation-mediated endocytosis, followed by cytoplasmic sorting, although the role of autophagy in this process is yet not clear [60]. 

Since chloroplast membrane proteins, such as translocon on the outer chloroplast membrane (TOC) proteins, can be ubiquitylated and degraded via the UPS [61], it is worth investigating how the UPS and selective autophagy coordinate the degradation process of intracellular membrane proteins. Considering that ≈80% of total ubiquitylated proteins are likely targeted by the 26S proteasome, we hypothesized that selective autophagy mediated degradation of ubiquitylated proteins is more limited to specific scenarios, such as membrane proteins or large protein complexes whose ubiquitylation is triggered by certain stress conditions. It is worth noting that the 26S proteasome itself, if damaged due to stress such as with MG132 treatment, is selectively targeted for autophagy degradation, suggesting a positive role of autophagy in regulating the function of the UPS [62]. We recently discovered that the 26S proteasome is targeted into the autophagy pathway through a new class of adaptor, called the ubiquitin-interacting motif (UIM)-docking site (UDS), dramatically increasing the pool of selective autophagy pathways [63]. It remains to be known whether any other UPS components are degraded via autophagy or which else stress conditions can trigger autophagy-mediated proteasome degradation.

SUMO, like Ub, is ligated to the lysine residues of a substrate resulting in mono- or multi-SUMOylation [64,65,66]. SUMO conjugation is far less complex but follows a similar cascade reaction to ubiquitylation (Figure 1) [54,64]. Interestingly, the ligation of SUMO onto its substrate can occur in the absence of a SUMO E3 ligase, and the SUMOylation sites commonly share a consensus pattern as part of a ΨKxD/E (Ψ = hydrophobic, x = any amino acid) motif recognized by a SUMO conjugation enzyme [67,68,69,70]. Hence, unlike the complex ubiquitylation profiles in which no consensus patterns were resolved [21], proteome-wide analysis revealed a significant enrichment of SUMOylated proteins involved in chromatin remodeling/repair, transcription, RNA metabolism, and protein trafficking [70]. The rapid increase of SUMOylated proteins upon various stresses suggested a substantial role of SUMO in general stress response [70,71]. Excellent reviews and perspectives regarding the biochemical process and biological functions of protein SUMOylation in plant stress response have been discussed elsewhere [65,66,72,73]. However, it is noteworthy to mention that in multiple pathways, SUMOylation and ubiquitylation intersect, demonstrating antagonistic, synergistic, and even more complex outcomes [74], further supporting the idea of convergent evolution between the ubiquitylation and the SUMOylation systems in plants.

Understanding the function and diversity of the UPS system is imperative to help dissect the vast number of pathways in which the system is involved. Over the past two decades, functional genomic studies have demonstrated that the UPS acted as both positive and negative regulators in plant stress response. Hereafter, we discuss such roles on the basis of recent studies on the diverse functions of 29 selected Ub ligases, 27 of which are E3 ligases, in abscisic acid (ABA) signaling, dehydration-responsive element binding 2A (DREB2A)-mediated stress tolerance, reactive oxygen species (ROS) homeostasis, and innate immune response (Table 2). 

## 4. Reinforcement Intersection of the UPS with ABA in Abiotic Stress Response

During stress response, the UPS acts as a crucial switch to constitutively or conditionally degrade activator/repressor to dampen/activate a stress response, respectively. This “switch” is, however, heavily dependent on a variety of factors. Generally, a stress is sensed, and that information is relayed by phytohormones [98]. In the case of drought, abscisic acid (ABA), has been revealed as a major phytohormone to activate drought resistance [99]. In addition to drought, ABA has been linked to abiotic stresses resulting from various unfavorable salinity and temperature conditions [99,100]. ABA also plays a vital role in seedling development and the flowering of plants [101]. The perception and transduction of stress is largely influenced by the UPS through regulating the ABA signaling [102]. Take, for example, the protein phosphatase 2C protein (PP2C) named ABA insensitive 1 (ABI1) (Figure 2) [103]. ABI1 and other PP2C proteins are believed to be stress signal transducers within the plant cell [104,105]. Dominant negative *abi1-1* mutant becomes insensitive to ABA, suggesting that ABI1 is a negative regulator of ABA signaling [75,105,106]. The E2 Ub conjugating enzyme 27 (UBC27) conjugates with ABI1 before bringing it to a RING E3 ligase, called ABA-insensitive RING protein 3 (AIRP3). ABI1 is ubiquitylated and marked for degradation, and thus UBC27 acts as a positive regulator of ABA signaling [75]. Interestingly, ABA activates the expression of *UBC27*, blocks UBC27 autoubiquitylation, and strengthens the interaction of UBC27 with ABI1, thus enhancing the role of UBC27 in ABA signaling [75]. It is known that one E3 ligase can target multiple substrates and vice versa [23]. In addition to the UBC27–AIRP3 pathway, ABI1 is also targeted by two U-box E3 ligases, PUB12 and PUB13, in an ABA receptor dependent manner (Figure 2) [78]. 

To date, a number of core proteins involved in ABA signaling have been identified as ubiquitylation substrates targeted by various types of E3 ligases, which act both negative and positive roles in ABA signaling. Such proteins include ABA receptors (pyrabactin resistance (PYR)/PYR1-like (PYL)/regulatory components of ABA receptor (RCAR) family proteins), co-receptor PP2Cs, sucrose non-fermenting-1 (SNF1)-related protein kinases (SnRK2s), and ABI5/ABF transcription factors. A detailed E3-substrate list on this topic has been reviewed elsewhere [102,107]. However, how the activities of E3 ligases correlate with ABA has largely remained a mystery. In addition to AIRP3, RING domain ligase 1 (RGLG1) has been discovered as a second RING E3 ligase that targets the degradation of PP2CA proteins, which is enhanced by ABA [108]. However, how ABA promotes RGLG1-mediated PP2CA turnover was unknown until one recent discovery showing the impact of ABA on the subcellular localization of RGLG1 and the nuclear interaction of RGLG1 with PP2CA (Figure 2) [109]. RGLG1 undergoes a posttranslational modification known as myristoylation in which a myristoyl group is added to alpha-amino group of an N-terminal glycine residue of the protein [110,111]. This modification changes the subcellular localization of RGLG1 to the plasma membrane. However, in the presence of ABA, RGLG1 myristoylation is compromised due to ABA suppression of N-myristoyltransferase 1 (NMT1). As a consequence, nonmyristoylated RGLG1 is translocated into the nucleus that facilitates its binding with PP2CA, thus promoting nuclear degradation of PP2CA and releasing SnRK2s for activating ABA responsive transcription factors [102,109]. Unlike the well-documented “molecular glue” function of auxin and jasmonic acid in SCF^TIR1/AFB^ and SCF^COI1^-mediated AUX/IAA and jasmonic acid receptor (JAZ) protein degradation [87,112,113,114,115,116], respectively, the new role of ABA in changing the intracellular localization of an E3 ligase has shed new insight into the intimate interaction between plant hormone and the UPS functions. 

In addition to intracellular shuttling, ABA can also directly affect the expression of an E3 ligase. A recently discovered E3 ligase, AtPPRT3, is directly affected by ABA concentrations within the cell [77]. Genomic analysis found ABA responsive elements at the promoter of *AtPPRT3*, and the expression of *AtPPRT3* transcripts is upregulated in the presence of exogenous ABA. These data are reinforced with knockouts and overexpression lines that become hyposensitive and hypersensitive to exogenous ABA, respectively. *AtPPRT3* could then work as a positive regulator of ABA to transduce the ABA signal to downstream pathways through potential interaction with transcription factors such as ABI3 or other ABA-dependent E3 ligases [77]. Considering the large transcriptome regulation by ABA and the important role of the UPS in ABA signaling, it is highly possible that other ABA-responsive E3 ligases would be further discovered to influence ABA-mediated stress responses.

## 5. Multiple UPS Routes Acting Upon DREB2A-Mediated Stress Tolerance

Multiple E3 ligases may act on the same pathway of a stress response or even to one common substrate to tune the response. Such activity is not necessarily in one direction, but rather works in a complex network to regulate the response in accordance with the strength of the stress. A well-documented process involved in such a response is exemplified by the degradation of DREB2A mediated by a RING E3 ligase, DREB2A-interacting protein 1/2 (DRIP1/2) (Figure 3) [84]. DREB2A is a transcription factor that has been linked as a positive regulator in dehydration, high salinity response, and thermotolerance [117,118,119]. Under normal conditions, DREB2A is present at very low concentrations and its transcripts are induced upon dehydration and high-salt stress [118]. Constitutive overexpression of DREB2A is counteracted by DRIP-mediated degradation, which prevents downstream expression of drought-related genes (Figure 3) [83,84]. DREB2A–DRIP-mediated drought responsive pathway seems conserved across evolutionarily distant plants. Recent studies in wheat (*Triticum aestivum*) shows that only when DRIP1/2 are degraded by a second E3 ligase, called *T. aestivum* stress-associated protein 5 (TaSAP5), are the levels of DREB2A allowed to increase. However, the increase of DREB2A is mild [83]. Additionally, knockouts of *DRIP* only partially allow for increased accumulation of DREB2A, suggesting that DREB2A is under additional regulatory elements [120]. 

A recently identified 30 amino acid negative regulatory domain (NRD) adjacent to the ERF/AP2 DNA-binding domain of DREB2A is the site of posttranslational modification. When this domain is deleted, DREB2A becomes stabilized and carries on its constitutive action in enhancing tolerance to drought and heat stresses [117,121]. Genetic and localization assays identified that a MATH domain-containing BTB protein (BPM) functions as a DREB2A receptor in a Cul3a/b-BTB E3 ligase. CRL3^BPM^ degrades DREB2A in an NRD-dependent manner. Biochemical data indicate that the NRD domain, by itself, is able to sufficiently interact with BPM. Expressing DREB2A missing the NRD allows for increased accumulation of DREB2A and activation of genes associated with drought response and thermotolerance [51]. In this instance, however, two separate E3 ligases control the abundance of DREB2A. Its stability and accumulation are negatively regulated by mono-subunit DRIP and multi-subunit CRL3^BPM^ RING E3s and positively regulated by a third mono-subunit TaSAP5 RING E3 enzyme.

To add further complexity, a new study recently identified a positive regulatory role of SUMO on the function of DREB2A. SUMOylation of an adjacent lysine to the NRD, K163, results in interference of BPM–DREB2A interaction, thus stabilizing DREB2A [122]. A large pool of SUMOylation substrates are identified upon heat shock treatment [70]. Interestingly, the stability of DREB2A is also increased by heat shock-induced SUMOylation of DREB2A, indicating a positive regulation of SUMO in heat tolerance [70,122]. SUMOylation of a potential target protein has been shown to suppress the function of ubiquitylation [123,124]. Hence, SUMOylation may tune the degradation of DREB2A in response to the abiotic stress (Figure 3). 

## 6. Influence of the UPS in Stress Regulation through ROS Homeostasis

Many E3 ligases that regulate abiotic stress in plants also influence ROS metabolism. ROS were once thought to be toxic byproducts of cellular respiration or photosynthesis due to over-accumulation of electrons. However, over the years new light has been shed on ROS as hormone-like molecules that the cell can utilize as ways to activate stress signaling [125,126]. Artificially upregulating or downregulating ROS within the cell can cause an equal opposite shift of ROS production to return to the basal level, suggesting that the production of ROS may be more intentional than a simple byproduct [127,128]. 

Over the last decade, many different E3 ligases have been identified, working in a manner that influences ROS production. Examples include AIRP1, AtL61, PUB19, and AtAIRP4, which all act in an ABA-mediated drought response [46,47,76,81]. However, the role of E3 ligases in crosstalk between ROS and ABA signaling has largely remained unknown [129]. One of the earliest signs of stomata movement is an increase accumulation of ROS within guard cells [130]. The process of achieving drought tolerance could potentially be ABA-mediated stomata closure through signal transduction of hydrogen peroxide (H_2_O_2_) involving calcium [131,132,133]. Stomata closure of the *atpub18-2atpub19-3* double knockout mutant showed a hypersensitive responsive to H_2_O_2_ but not to calcium, suggesting that AtPUB18 and AtPUB19 are involved in ABA signaling pathway downstream of H_2_O_2_ and upstream of calcium [80].

The activity of a *Populus euphratica* ubiquitin E3 ligase named PeCHYR1 has been recently discovered to correlate with ROS production. Overexpressing *PeCHYR1* resulted in increased sensitivity to exogenous ABA, elevated levels of ROS, enhanced stomata closure, and improved water retention [86]. One potential target of PeCHYR1 for increasing ROS may be protein phosphatase 2A (PP2A), an enzyme involved in ROS signaling [134]. This phosphatase has been discovered to undergo non-proteolytic ubiquitylation by AtCHIP E3 ligases, thus increasing its activity in regulating ROS metabolism (Figure 4). Similar to PeCHYR1, overexpression of AtCHIP upregulated the plant sensitivity to exogenous ABA treatment [85]. 

ROS products in chloroplasts and mitochondria have been recognized as one type of retrograde signal to coordinate both organelle and nuclear genome activities. Due to highly energetic electron transport reactions in both organelles, the redox status of their elements is easily perturbed by stress, making them key cellular sensors of environmental fluctuations. Emerging evidence has suggested that the UPS is also involved in selective turnover of both membrane and lumen chloroplast proteins, thus regulating the chloroplast activity and, in some cases, facilitating the recycling of damaged chloroplasts through autophagy [61,135,136]. For example, chloroplast biogenesis requires precise quality control of TOC protein complex. TOC facilitates the translocation of chloroplast components necessary for chloroplast function. In the event of abiotic stress, translocation can be aborted via TOC degradation by a RING E3 ligase, suppressor of PPI1 locus1 (SP1) (Figure 4) [61]. Later studies from the same group also demonstrated that such regulation increases stress tolerance of plants by reducing plastid import of photosynthesis proteins to avoid oxidative stress [137]. However, how a stress signal triggers the regulation of UPS in chloroplast function remains further investigation. Given the role of ABA in stress response and its crosstalk with H_2_O_2_, we speculate that both ABA and H_2_O_2_ could act either upstream or downstream of the UPS-mediated quality control of chloroplasts upon stress treatment.

## 7. Role of the UPS in Innate Immune Response 

Similar to animals, plants utilize innate immunity to respond to pathogen challenges [138,139]. However, unlike animals, plants lack adaptive immunity, whose response is exacerbated and more specific to the pathogen following a previous infection [140,141,142]. Instead, plants rely on a suite of sophisticated innate immune responses that generally result in two outcomes: systemic acquired resistance (SAR) and hypersensitive response (HR). To date, it has been recognized that nearly every step from pathogen recognition to response is under UPS-mediated regulation. Plants adopted the UPS to act as a quick switch to activate pathogen-related (PR) proteins and induce salicylic acid (SA)-mediated SAR [17,143,144,145,146]. Control of *PR* gene expression can be a delicate dance within the cell. The seesaw between growth and pathogen resistance is juggled to ensure proper development. Activation of *PR* genes prematurely may result in delayed growth and compromised propagation [99]. To properly time this response is balanced by hormones such as SA, jasmonic acid (JA), and ethylene [147], whose functions are intimately connected with the UPS [3,146,148,149]. 

Detection of pathogens can take two separate routes: pathogen-associated molecular patterns (PAMP)-triggered immunity (PTI) or effector-triggered immunity (ETI), which directly detect pathogen proteins such as the flagellum or effector (avirulence) proteins released by a pathogen, respectively [150,151,152]. The plant immune system has developed a checkpoint regulatory route to maintain a homeostasis of these receptors to effectively respond to pathogens without causing an autoimmune response (Figure 5). For example, PTI receptors, such as flagellin-sensitive 2 (FLS2) and lysm-containing receptor-like kinase 5 (LYK5), are constantly degraded by a PUB13 E3 ligase in normal growth conditions to prevent constitutive immune response [79,153]. If a pathogen is able to avoid PTI, ETI receptors (resistance (R) proteins) may take action to function as the second defense barrier to pathogens [152]. One large group of these receptors are attributed to the nucleotide-binding leucine rich repeat (NLR) protein family [41,154,155,156]. An F-box protein, termed constitutive expressor of pathogenesis-related genes 1 (CPR1), along with an E4 ubiquitin-conjugating factor, MUSE3 (for mutant, snc1-enhancing 3), negatively regulates the NLR receptor suppressor of NPR1 constitutive 1 (SNC1) [88,91,157]. 

Genetic screens have uncovered multiple redundant SNC1-like receptors that play similar roles and are under the same negative regulation as SNC1 [82]. These receptors, named sidekick SNC1 1 (SIKIC1), associate with SNC1 to regulate its function. To suppress autoimmunity, SIKIC1 is under the control of two RING E3 ligases—MUSE1 and 2 [82]. It is yet unclear as to the mechanisms that are undergone to activate SNC1; however, what is clear is that there is an extensive network of regulatory E3 ligases that act to repress constitutive immune response. 

Conversely, the function of E3 ligases needs to be tuned upon immune response. In addition to previous discovery of an F-box protein, coronatine insensitive 1 (COI1), which functions as JA receptor, recent discoveries demonstrated that a BTB protein, nonexpressor of pathogenesis-related genes 1 (NPR1), is a SA receptor [94,95]. When mutated, *npr1* plants have an underwhelming response to pathogen attacks with reduced PR expression, suggesting that *NPR1* is a positive regulator of immune response [92]. In contrast, two paralogs of *NPR1, NPR3*, and *NPR4*, negatively regulate pathogen response [40,95]. Both *npr3* and *npr4* mutants display increased PR expression and increased resistance to disease. This phenotype is exacerbated with *npr3 npr4* double mutants. NPR3 and NPR4 are believed to act as SA co-receptors, functioning as CRL3 E3 ligases. Under normal conditions, NPR3 and NPR4 bind and degrade NPR1 [95]. However, once a pathogen is sensed through ETI, upregulated SA binds to NPR4 and causes a major conformational remodeling in its SA-binding core, thus weakening its interaction with NPR1 and activating NPR1-mediated positive immune response [94]. Therefore, SA not only acts as a repressor of NPR4–NPR1 interaction but also promotes the binding of NPR1 with nuclear genome DNA where NPR1 acts as a transcriptional activator of pathogen-responsive genes [40,93].

ROS metabolism is connected to nearly all kinds of stress response. NADPH oxidase-mediated accumulation of extracellular ROS is known as one immune response conserved in both plant and animal cells [158]. Recent studies also discovered the involvement of ubiquitylation in this process (Figure 5). Under normal conditions, plasm membrane-localized NADPH oxidase, respiratory burst oxidase homolog protein D (RBOHD), is phosphorylated at its C terminus by AvrPphB susceptible1-like13 (PBL13) kinase [159]. Such phosphorylation allowed constitutive turnover of RBOHD through ubiquitylation-mediated endocytosis by PBL13 interacting RING domain E3 ligase (PIRE), thus keeping RBOHD activity at a basal level. Consequently, *pire* mutants treated with bacterial flagellin (flg22) have increasing ROS burst and accumulation of RBOHD, confirming that PIRE negatively regulates RBOHD-mediated innate immune response. Conversely, PIRE itself becomes heavily phosphorylated by an unknown kinase following RBOHD-activated ROS burst and immune activation [96]. This intricate system involving phosphorylation and ubiquitylation provides the plant with multitude levels of regulation of the immune response.

The UPS also influences plant HR. As discussed above, chloroplasts, such as mitochondria, are plant organelles that play important roles in generating ROS, which can lead to HR-like cell death mediated by a mitogen activated kinase cascade [160]. The role of UPS in regulating chloroplast ROS homeostasis implies a possibility of UPS-mediated HR. In addition, the UPS also directly regulates the stability of transcription factors involved in HR. One pathway is exemplified by MYB domain protein 30 (MYB30)-mediated HR pathogen defense (Figure 6). MYB30 is a transcription factor that activates expression of genes associated with stress including those involved in very long fatty acid biosynthesis [161,162]. During standard growth conditions, MYB30 is expressed and marked for degradation by a RING E3 ligase MYB30-interacting E3 ligase1 (MIEL1) to prevent constitutive activation of HR [90]. It was discovered that MIEL1 itself is also subjected to degradation in a UPS-dependent manner, likely through autoubiquitylation, which is prevented by the interaction of phosphatidylinositol 4-kinase γ2 (PI4Kγ2). Dissociation between PI4Kγ2 and MIEL1 promotes MIEL1 autoubiquitylation and turnover. As a consequence, MYB30 is stabilized to activate HR reactions [163]. However, how PI4Kγ2 is prevented from interacting with MIEL1 remains further investigation. One possibility could involve ABA signaling [164]. Interestingly, MIEL1 also targets the degradation of MYB96, a transcription regulator of ABA signaling [164]. The research found that both ABA treatment and pathogen infection enhanced accumulation of MYB96 and expression of MYB30-responsive genes by promoting MIEL1 turnover, suggesting that MIEL1 bridges the interplay between ABA and biotic stress signaling [164]. In addition to MIEL1, a RING E3 ligase, RING-H2 finger protein 2B (RHA2b), also interacts and degrades MYB30 in an ABA-dependent manner during drought response [97]. Whether RHA2b, like MIEL1, regulates plant HR is yet unknown. 

## 8. Genome Evolution of the Plant UPS and Future Perspective

The UPS is one of the largest regulatory systems in plant cells. The ability of rapidly degrading regulators to turn on or off a metabolic pathway makes this system the best candidate for combating environmental challenges. However, converse to the large group of UPS genes discovered in plant genomes (Table 1) [5,22], only a handful number of UPS members, particularly the E3 ligase genes, have been functionally characterized [3,149]. For example, only 83 out of ≈800 predicted Arabidopsis *F-box* genes have had known functions to date [165]. Although it has been hypothesized that the large expansion of UPS in plants provides an essential genomic resource for evolutionary innovation, it remains elusive as to why the functions of so many predicted E3 ligase genes are unknown. 

Functional redundancy due to high rate of gene duplications, low abundance of both E3 and ubiquitylation substrates, short period when a particular UPS pathway is effective, challenges in identifying pairwise relationship between an E3 ligase and its substrate(s), and promiscuous functions of core UPS components could all contribute to the hardship of functional genomic studies in this field [3]. In addition to these challenges, we hypothesize that an inherent evolutionary mechanism attributed to the extensive expansion of the E3 ligase group could be a hurdle as well. Thanks to the advances of both genome and proteome sequencing technologies, innovative approaches utilizing multiple data resources could help overcome this difficulty. Here, we used the *F-box* gene family as an example to illustrate future directions in tackling the role of the large group of UPS genes in plant growth and development, particularly in stress response.

Through genome-wide comparison and phylogenetic analysis, we discovered a novel evolutionary mechanism in the *F-box* gene family whose members comprise the largest group of CRL E3 ligases. On the basis of the weak correlation of the size of the *F-box* family with the genome complexity and the large proportion of silenced *F-box* genes that are under epigenomic suppression, we hypothesized that the plant *F-box* gene family is under a genomic drift evolution [166,167]. Recently, we expanded our evolutionary studies of the *F-box* gene family from previous 18 plant genomes to 111 genomes that allowed us to model the distribution patterns of *F-box* genes [23]. Four clusters of *F-box* genes were discovered to be significantly distinct in evolutionary conservation and duplication patterns. While the largest cluster is lineage-specific and on average only present in 3 out of 111 genomes, the second largest cluster is specific to Brassicales. Statistical modeling suggests that the distribution of these two clusters of *F-box* genes is not a stochastic event. In addition to our unpublished data showing that overexpression of an *F-box* gene often results in a detrimental phenotype, we proposed a purifying selection model to explain the genomic drift evolution of these two clusters of *F-box* genes [23]. Because of the deleterious function, these two clusters of *F-box* genes are only limited in a small group of plants. Only when the activity of the *F-box* genes is suppressed, such as epigenomic suppression in Arabidopsis, would they be able to significantly duplicate in a plant genome [23]. Therefore, the size of the *F-box* gene family is not determined by the genome complexity but the presence of a mechanism that can suppress the activity of *F-box* genes. 

However, like genetic drifted alleles, some genomic drifted *F-box* genes may be activated to play adaptive roles through evolutionary innovation [23]. These members likely play important roles in stress response. But, how are we able to discover these genes? To tackle this question, we also recently developed a machine learning approach to prioritize the active *F-box* members in Arabidopsis. On the basis of genomic, expression, and evolutionary features of the 83 known Arabidopsis *F-box* genes, we utilized a neural network machine learning approach to predict a handful number of unknown *F-box* genes to be active with high confidence [165]. Thus, through iterative predictions based on new *F-box* genes characterized, we would be able to effectively identify the functions of active *F-box* gene members rather than being trapped in a large group of inactive ones. Such an integrative data analysis approach could be also adopted into the functional genomic studies of several other UPS families, in which many members are yet uncharacterized.

## Figures and Tables

**Figure 1 plants-10-00246-f001:**
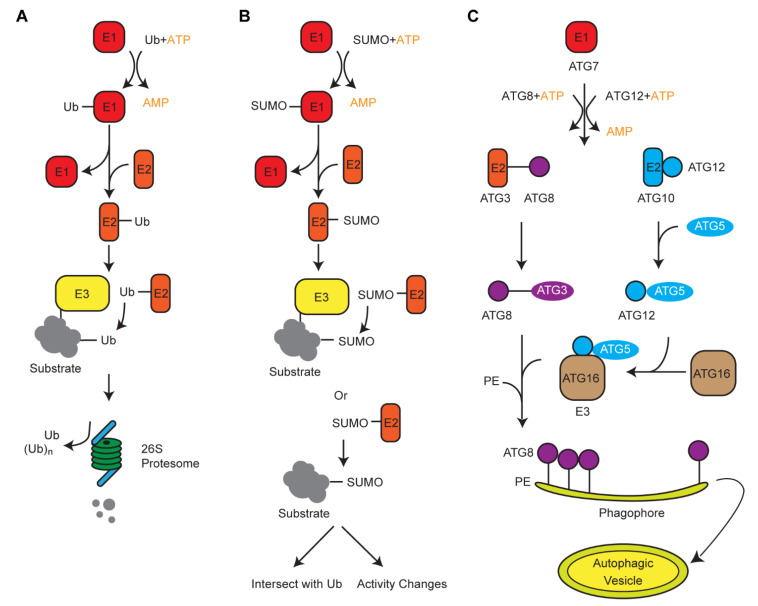
Schematic diagrams showing the ligation process of ubiquitin (Ub), small ubiquitin like-modifier (SUMO), and autophagy-related protein 8 (ATG8). (**A**,**B**) The ubiquitylation and SUMOylation processes use a similar cascade reaction to add Ub or SUMO moieties, respectively, on to a target protein. However, different E1, E2, and E3 enzymes are involved. Ubiquitylation utilizes a much larger set of E3 enzymes to target a diverse group of substrates. SUMOylation can also occur in the absence of E3. (**C**) Autophagy formation is mediated by a variety of ATG proteins that are similar to E1, E2, and E3 that act sequentially. Activation of ATG8 and ATG12 require an ATP-dependent ATG7. ATG8 and ATG12 then conjugate with ATG3 and ATG5, respectively. With the help of an E3 ligase ATG5–ATG12/16 complex, ATG8 conjugates with phosphatidylethanolamine to form a double-membrane autophagic vesicle, i.e., autophagosome, in which substrates are engulfed either non-selectively or selectively through interacting with ATG8.

**Figure 2 plants-10-00246-f002:**
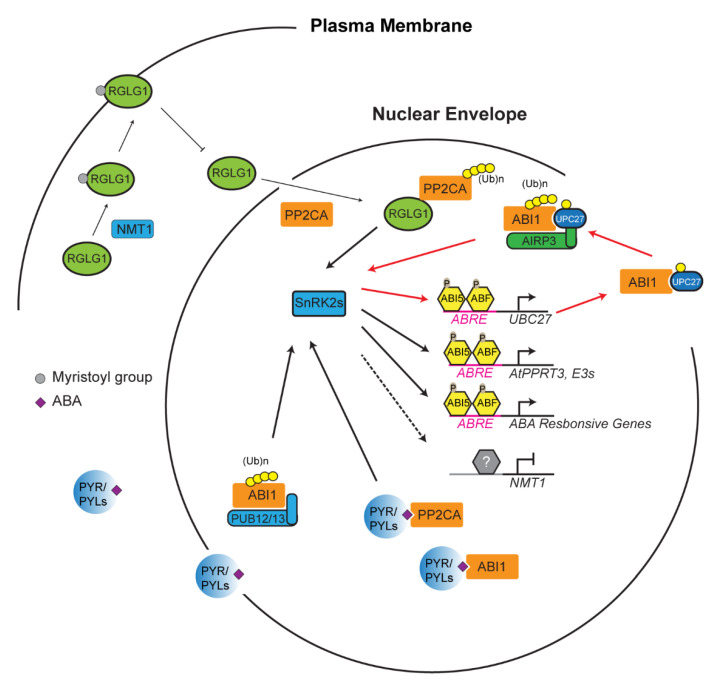
Ubiquitin-26S proteasome system (UPS)-mediated turnover of PP2CA promotes abscisic acid (ABA) signaling. PP2CA proteins including ABA insensitive 1 (ABI1) are ABA co-receptors that negatively regulate ABA response by inhibiting the phosphorylation activity of sucrose non-fermenting-1 (SNF1)-related protein kinases (SnRK2s). SnRK2s activities are activated either through its dissociation with PP2CAs that are competed out by ABA receptors, pyrabactin resistance (PYR)/PYR1-like (PYL), upon ABA perception or by UPS-mediated degradation of PP2CAs. Ubiquitylation of PP2CAs by really interesting new gene (RING) domain ligase 1 (RGLG1), ABA-insensitive RING protein 3 (AIRP3), and plant U-box (PUB)12/13 results in their degradation in the 26S proteasome, thus freeing SnRK2 proteins to activate ABI5 and ABF bZIP transcription factors. Expression upregulation of E2 (e.g., Ub conjugating enzyme 27 (*UBC27*)) and E3 ligases (e.g., *AtPPRT3*) and inhibition of N-myristoyltransferase 1 (*NMT1*) reinforce the turnover process of PP2CAs, thus promoting ABA signaling. Red lines highlight the reinforcement intersection between ABA signaling and UBC27–AIRP3-mediated ABI1 degradation. The latter process could happen in both cytoplasm and nucleus.

**Figure 3 plants-10-00246-f003:**
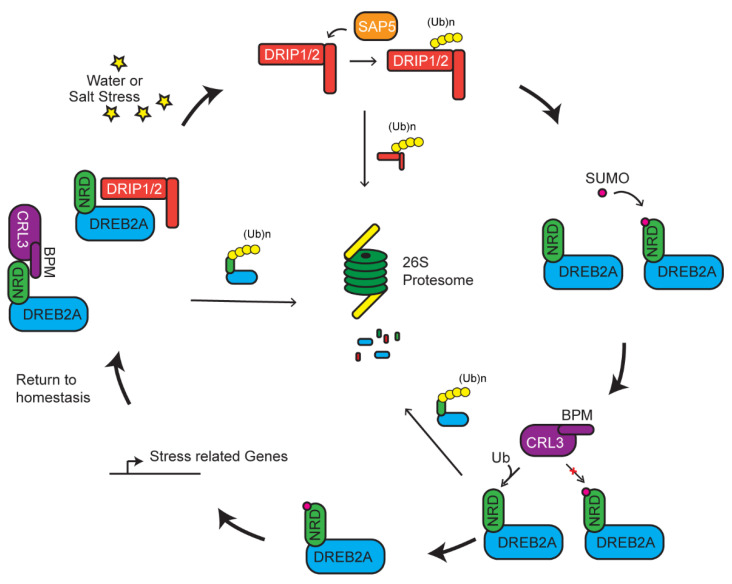
Multiple levels of regulation on dehydration-responsive element binding 2A (DREB2A) activity. Under normal conditions, DREB2A is targeted by CLR3^BPM^ and DREB2A-interacting protein 1/2 (DRIP1/2) for ubiquitylation and degradation. Presence of stress induces the degradation of DRIP1/2 by stress-associated protein 5 (SAP5)-mediated ubiquitylation. DREB2A is stabilized and further protected by SUMOylation that prevents it from CLR3^BPM^-mediated ubiquitylation. Accumulation of DREB2A activates stress related gene expression to mitigate the stress.

**Figure 4 plants-10-00246-f004:**
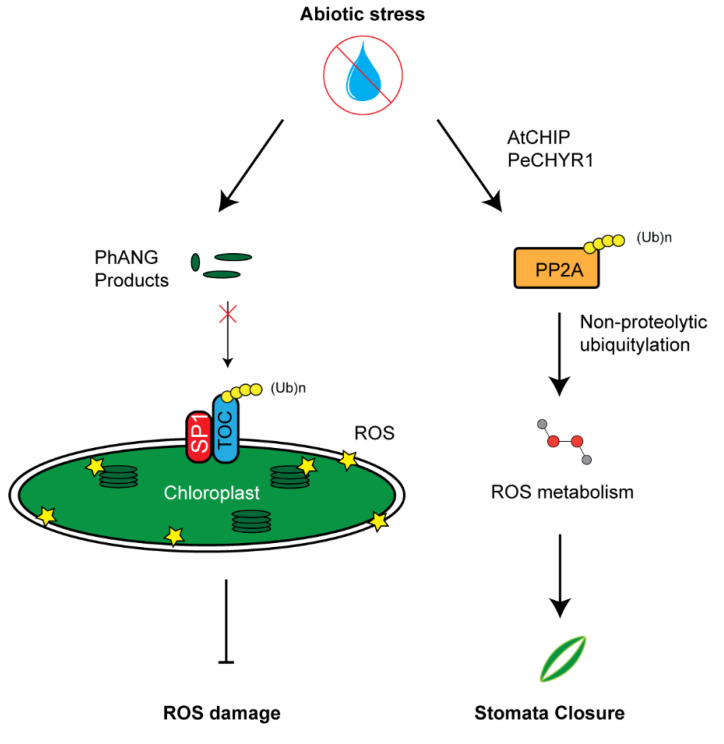
UPS-mediated reactive oxygen species (ROS) metabolism. **Left panel**: Chloroplast development and function needs to tightly coordinate with a large group of photosynthesis-associated nuclear genes (*PhANGs*) whose products are translocated into chloroplasts primarily through the gate of TOC complex. Upon stress, dysfunction of chloroplast can build high concentrations of ROS products. Constant import of PhANG products accelerates ROS production. SP1-medaited TOC33 and TOC159 degradation inhibits the import of PhANG products, thus reducing further chloroplast ROS damage. **Right panel**: AtCHIP/PeCHYR1 increases PP2A enzymatic function in ROS biosynthesis through non-proteolytic ubiquitylation. High concentration of ROS promotes stomata closure.

**Figure 5 plants-10-00246-f005:**
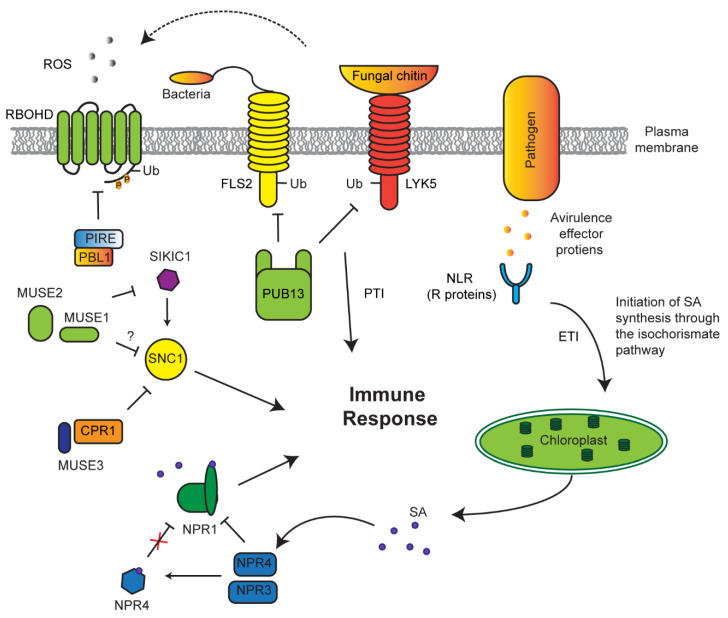
UPS regulation of innate immune response. The abundance of pathogen-associated molecular patterns (PAMP)-triggered immunity (PTI) receptors flagellin-sensitive 2 (FLS2) and lysm-containing receptor-like kinase 5 (LYK5) are under the control of E3 ligase plant U-box (PUB)13, which marks both for degradation. Avirulence effectors (AVR) expressed and released from pathogens into the cell perturbs native proteins and are recognized by host R proteins, such as nucleotide-binding leucine rich repeat (NLR). AVR–R interactions develops an effector-triggered immunity (ETI) signaling cascade to increase salicylic acid (SA) biosynthesis. Increased SA facilitates its binding with NPR4, resulting in the conformational changes of NPR4, which otherwise constantly interacts and degrades NPR1. NPR1, a positive regulator of immune response, is activated upon binding of SA. Suppressor of NPR1 constitutive 1 (SNC1) is one type of NLR that triggers constitutive innate immune response. To prevent its growth suppression, SCF^CPR1^ targets SNC1 for ubiquitylation and degradation under normal conditions in a manner enhanced by mutant, snc1-enhancing 3 (MUSE3) E4 ubiquitylation enzyme. A second type of NLR, sidekick SNC1 1 (SIKIC1), promotes SNC1-mediated immune response, whose function is controlled by two RING E3 ligases, MUSE1/2. PTI-promoted generation of apoplastic ROS via respiratory burst oxidase homolog protein D (RBOHD) is suppressed by AvrPphB susceptible1-like13 (PBL13) interacting RING domain E3 ligase (PIRE)-mediated ubiquitylation of phosphorylated RBOHD by PBL1 followed by endocytosis, cytoplasmic sorting, and vacuolar degradation.

**Figure 6 plants-10-00246-f006:**
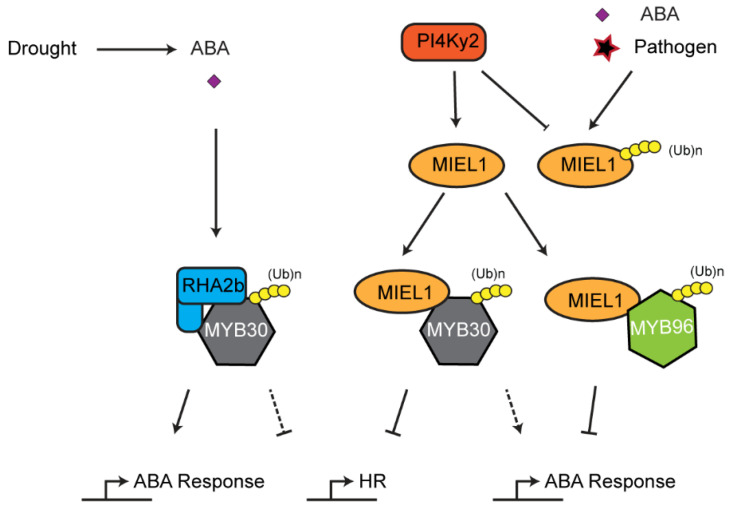
Interplay between ABA signaling and hypersensitive response (HR)-mediated immune response. MYB30 and MYB96 are two key transcription factors involved in HR and ABA signaling, respectively. Interaction between phosphatidylinositol 4-kinase γ2 (PI4Kγ2) and MYB30-interacting E3 ligase1 (MIEL1) stabilizes and promotes the ubiquitylation activity of MIEL1 on both MYB30 and MYB96, whereas both ABA treatment and pathogen challenge accelerate turnover of MIEL1 through autoubiquitylation. MYB30 is also targeted for ubiquitylation and degradation by another RING E3 ligase, RING-H2 finger protein 2B (RHA2b), upon drought-induced ABA signaling.

**Table 1 plants-10-00246-t001:** Number of E3 ligases predicted in 14 selected plant genomes [22].

Species	*BTB*	*F-box **	*HECT*	*RING ***	Sum
*Arabidopsis halleri*	63	850	9	480	1402
*Arabidopsis lyrate*	77	989	10	542	1618
*Arabidopsis thaliana*	64	697	7	516	1284
*Amborella tricopoda*	56	230	7	314	607
*Brachypodium distachyon*	178	813	10	554	1555
*Brassica rapa*	97	975	10	802	1884
*Boechera stricta*	75	505	9	473	1062
*Capsella rubella*	69	970	8	527	1574
*Leersia perrieri*	110	542	9	460	1121
*Oryza brachyantha*	78	264	12	391	745
*Oryza punctata*	111	535	8	485	1139
*Oryza sativa*	156	732	8	532	1428
*Sorghum bicolor*	158	678	9	563	1408
*Zea mays*	95	331	20	699	1145

* Only two F-box subfamilies, F-box and F-box-like, were analyzed. Three rare F-box subfamilies, F-box-like_2, F-box_4, and F-box_5, were not included but have been added in a recent study by Hua [23]. ** The number indicates the group of mono-subunit RING E3s. Abbreviations: BTB: Broad complex-Tramtrack-Bric a brac; HECT: Homologous to the E6-AP Carboxyl Terminus; RING: Really Interesting New Gene.

**Table 2 plants-10-00246-t002:** List of Ub ligases discussed in this work.

Protein	Accession	Involved Stress Pathway (Substrates)	Protein Family	Reference
**ABA Signaling**				
AIRP1	AT4G23450.2	Positive regulator in ABA-mediated drought tolerance	RING E3 ligase	[46]
AIRP3	AT3G09770.1	E3 ligase for ABI1 degradation (ABI1)	RING E3 ligase	[75]
AtAIRP4	AT2G36270.1	Positive regulator in ABA-mediated drought tolerance	RING E3 ligase	[76]
AtL61	AT3G14320.1	Positive regulator in ABA-mediated drought tolerance	RING-H2 E3 ligase	[47]
AtPPRt3	AT1G18470.1	Positive regulator in ABA-mediated drought tolerance	RING E3 ligase	[77]
PUB12	AT2G28830.1	Drought and immune response (ABI1)	U-box E3 ligase	[78]
PUB13	AT3G46510.1	Drought and immune response (ABI1)	U-box E3 ligase	[79]
PUB18	AT1G10560.1	ABA signaling	U Box E3 ligase	[80]
PUB19	AT1G60190.1	Negative regulator of ABA-mediated drought tolerance	U-box E3 ligase	[81]
RGLG1	AT3G01650.1	Positive regulator of ABA-mediated drought tolerance (PP2CA)	RING E3 ligase	[82]
UBC27	AT5G50870.2	E2 that conjugates with ABI1	E2	[75]
**DREB2A-Mediated Stress Tolerance**				
BPM	AT1G21780.1	Thermotolerance (DREB2A)	CRL-BTB E3 ligase	[51]
SAP5	AT3G12630.1	Thermotolerance (DRIP1/2)	A20/AN1 domain containing E3 ligase	[83]
DRIP1/2	AT1G06770.1	Thermotolerance (DREB2A)	RING E3 ligase	[84]
**ROS Homeostasis**				
AtCHIP	AT3G07370.1	Regulator of temperature response (PP2A)	U Box E3 ligase	[85]
PeCHYR1	AT5G22920.1	Positive regulator in ABA-mediated drought tolerance	Ring E3 ligase	[86]
SP1	AT1G63900.2	Chloroplast biogenesis and ROS homeostasis (TOC complex)	RING E3 ligase	[61]
**Innate Immune Response**				
COI1	AT2G39940.1	Jasmonic acid receptor (JAZ proteins)	SCF CRL E3 ligase	[87]
CPR1	AT4G12560.1	Negative regulator of immunity (SNC1)	SCF CRL E3 ligase	[88,89]
MIEL1	AT5G18650.1	Immune response (MYB30)	RING E3 ligase	[90]
MUSE1	AT3G58030.1	Regulates SIKIC1 (SIKIC1)	RING E3 ligase	[82]
MUSE2	AT2G42030.1	Regulates SIKIC1 (SIKIC1)	RING E3 ligase	[82]
MUSE3	AT5G15400.1	Mediates SNC1 ubiquitylation	E4 ligase	[91]
NPR1	AT1G64280.1	SA receptor	CRL-BTB E3 ligase	[92,93,94]
NPR3/4	AT5G45110.1	SA co-receptor (NPR1)	CRL-BTB E3 ligase	[95]
PIRE		RBOHD degradation (RBOHD)	RING E3 ligase	[96]
PUB12	AT2G28830.1	Drought and immune response (FLS2)	U-box E3 ligase	[78]
PUB13	AT3G46510.1	Drought and immune response (LYK5)	U-box E3 ligase	[79]
RHA2B	AT2G01150.1	Immune response (MYB30)	RING E3 ligase	[97]

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
