# Peer review of "The Ubiquitin Switch in Plant Stress Response"

_plants, 2021, doi:10.3390/plants10020246_

Round 1
Reviewer 1 Report
The authors make an attempt to show the subject in a slightly different way than other reviews. Essentially, the review contains numerous valuable parts. Yet, it confused me in several places.
1) General comment: I do not like the idea of linking the “ubiquitin switch” only to the ubiquitin-26S proteasome system (UPS). The authors should make sure that selective autophagy of ubiquitinated targets is sufficiently clarified in the review. In the chapter ‘Protein turnover by the plant UPS’ the authors (a sort of) make the statement that more than 80% of the total ubiquitinated proteins are degraded by the 26S proteasome and provide ‘unpublished data’ as a reference for this statement. Nevertheless, they should pay more attention to the autophagy process because it is also linked to Ub via receptors such as NBR1. Then, the fact that E3 targets the protein does not mean that this protein is targeted to UPS, unless it is proved that it is indeed degraded via 26S proteasome. I find somewhat misleading the lack of clear information that selective autophagy is responsible for selective degradation of Ub-tagged proteins.
2) There is a clash between the groups of E3 ligases shown in Table 1 and the groups of E3 ligases described on page 3 below Table 1. In the text, PUB, HECT, RING of E3 ligases are mentioned as three main types, while the table contains BTB, F-box, HECT, and RING. It should be somehow coordinated. And also, I do not see in the Table the support for the statement ‘Our recent studies suggest that >1,000 RING E3 ligase genes are encoded in both monocotyl and dioctyl plant genomes (Table 1; [22]).’- I mean that in Table 1 no plant has >1000 RING-type E3 ligases. Besides, the authors might consider removing Table 1 and the pertinent part of the text.
3) Fig.1 is fine and generally, the part of the text comparing Ub, ATG8, and SUMO is clear. But then is Table 2 which is not sufficiently linked to this chapter and it is limited only to the E1-E3 involved in ubiquitination. Besides, the alphabetic order of ligases in this table is confusing. It would e better to organize them according to the pathway, especially if the authors could add the targets. There is one typing error (in the line of AtCHIP) in Table 2. Besides, I would rather say that RGLG1 is a positive factor of ABA signaling (it releases the blockade of ABA signaling by PP2C by degradation PP2C).
4) on page 7 the authors state that ‘...myristoylated RGLG1 is translocated into nucleus which facilitates its binding with PP2CA transcription factors [89].’ I am not aware that PP2C acts as a transcription factor. To my knowledge, it is a phosphatase that negatively regulates ABA signaling via blocking SnRK2s. Please correct. Besides, would it be possible to add a figure clarifying information provided in the chapter about intersection UPS-ABA?
5) The same comment (if possible provide the figure) is to the chapter about the UPS-ROS interplay. Besides, explain TOC.
6) I like Figure 3 and the pertinent text but then, there is a part of the text about plant HR (please check also if HR is explained) that is not sufficiently linked to the previous part
7) The text in the last chapter is difficult to follow. Partially because of the not precise language. For example, what the authors mean by ‘the number of UPS loci’? Besides, I do not understand why the main focus of the last chapter of this otherwise is a general review is on F-box proteins.
Author Response
Response to Reviewer1’s Comments
Comment 1: The authors make an attempt to show the subject in a slightly different way than other reviews. Essentially, the review contains numerous valuable parts. Yet, it confused me in several places.
Response 1: We appreciate the positive comments received from this reviewer. It was our best interest to provide new insight into the field. We apologize for the confusion raised. We hope that this reviewer finds the revisions satisfactory with all the comments addressed.
Comment 2: General comment: I do not like the idea of linking the “ubiquitin switch” only to the ubiquitin-26S proteasome system (UPS). The authors should make sure that selective autophagy of ubiquitinated targets is sufficiently clarified in the review. In the chapter ‘Protein turnover by the plant UPS’ the authors (a sort of) make the statement that more than 80% of the total ubiquitinated proteins are degraded by the 26S proteasome and provide ‘unpublished data’ as a reference for this statement. Nevertheless, they should pay more attention to the autophagy process because it is also linked to Ub via receptors such as NBR1. Then, the fact that E3 targets the protein does not mean that this protein is targeted to UPS, unless it is proved that it is indeed degraded via 26S proteasome. I find somewhat misleading the lack of clear information that selective autophagy is responsible for selective degradation of Ub-tagged proteins.
Response 2: Thanks for pointing out the role of selective degradation of Ub-tagged proteins via autophagy. We did recognize the role of autophagy in protein degradation. However, we believe that the UPS and autophagy most likely target a different set of ubiquitylated proteins. In general, autophagy mediates the degradation of large protein complexes, such as proteasome, rather than individual proteins. Since autophagy is not the primary topic of this manuscript, we suggested the readers to read other related articles in our original submission, in which we stated, “…, the role of autophagy in stress regulation and organelle recycling has been reviewed elsewhere [55-57]” (Line 172 -173).
However, recent studies in yeast found that ubiquitylated vacuole membrane proteins are targeted for degradation via autophagy, which raised an open question whether this is also the case in plant cells. Since chloroplast membrane proteins, such as TOC proteins, can be ubiquitylated and degraded via the UPS, it is worth investigating how the UPS and selective autophagy coordinate the degradation process of membrane proteins. Once again, we appreciate this reviewer’s recognition about our unpublished data showing “more than 80% of the total ubiquitinated proteins are degraded by the 26S proteasome”. We hypothesized that even though some ubiquitylated proteins could be degraded via autophagy, the group of these proteins is likely limited to specific scenarios, such as membrane proteins or large protein complexes whose ubiquitylation is triggered by certain stress conditions. We have added this new information in the revision (Lines 173-188)
Comment 3: There is a clash between the groups of E3 ligases shown in Table 1 and the groups of E3 ligases described on page 3 below Table 1. In the text, PUB, HECT, RING of E3 ligases are mentioned as three main types, while the table contains BTB, F-box, HECT, and RING. It should be somehow coordinated. And also, I do not see in the Table the support for the statement ‘Our recent studies suggest that >1,000 RING E3 ligase genes are encoded in both monocotyl and dioctyl plant genomes (Table 1; [22]).’- I mean that in Table 1 no plant has >1000 RING-type E3 ligases. Besides, the authors might consider removing Table 1 and the pertinent part of the text.
Response 3: We apologize for this confusion. It was our oversight that did not clearly address that the RING E3s in Table 1 indicated the subgroup of mono-subunit RING E3s. We have added this note in Table 1 in the revision. As we stated in the original submission, the entire group of RING E3s is composed of both mono-subunit and multi-subunit E3 enzymes. The sum of number of BTB-type CRL, F-box-containing CRL, and mono-subunit RING E3s in 12 out of 14 plant genomes shown in Table 1 is > 1,000. We think the sheer number of E3 ligases indirectly suggests the diverse role of UPS in plant stress response, which is also partly suggested by Reviewer 2. We prefer to keep Table 1.
Comment 4: Fig.1 is fine and generally, the part of the text comparing Ub, ATG8, and SUMO is clear. But then is Table 2 which is not sufficiently linked to this chapter and it is limited only to the E1-E3 involved in ubiquitination.
Response 4: We added the general pathway comparison among Ub, ATG8, and SUMO in order to provide the readers with a broad view about the role of these three small proteins in protein turnover and their potential crosstalk, which has been discussed in the section of “Crosstalk of Ub with Its Two Siblings, ATG8 and SUMO, in Protein Turnover”. It would make it lengthy and hard to read if we were to discuss the stress responses involving all the three small proteins. Therefore, the focus of this manuscript is to discuss the role of UPS in plant stress response. The roles of autophagy and SUMO have been recommended to read other excellent reviews in our original submission in Lines 173 and 221, respectively.
Comment 5: Besides, the alphabetic order of ligases in this table is confusing. It would e better to organize them according to the pathway, especially if the authors could add the targets. There is one typing error (in the line of AtCHIP) in Table 2.
Response 5: Thanks for the suggestion. We have revised the table based on the four topics/pathways discussed in the manuscript. We have also corrected the typos herein and elsewhere in the manuscript.
Comment 6: Besides, I would rather say that RGLG1 is a positive factor of ABA signaling (it releases the blockade of ABA signaling by PP2C by degradation PP2C).
Response 6: We really appreciate this good catch. It was our mistake and we have corrected it in the revised Table 2.
Comment 7: on page 7 the authors state that ‘...myristoylated RGLG1 is translocated into nucleus which facilitates its binding with PP2CA transcription factors [89].’ I am not aware that PP2C acts as a transcription factor. To my knowledge, it is a phosphatase that negatively regulates ABA signaling via blocking SnRK2s. Please correct.
Response 7: We really appreciate this reviewer’s carefully reading. It was our mistake. We have corrected this sentence in the revision that now reads, “As a consequence, nonmyristoylated RGLG1 is translocated into nucleus that facilitates its binding with PP2CA, thus promoting nuclear-degradation of PP2CA and releasing SnRK2s for activating ABA responsive transcription factors [100, 107].” (Lines 319-322)
Comment 8: Besides, would it be possible to add a figure clarifying information provided in the chapter about intersection UPS-ABA?
Response 8: Thanks for this excellent suggestion. We have developed a new figure (Figure 2) to illustrate the reinforcement intersection between ABA signaling and UPS-mediated turnover of PP2CAs.
Comment 9: The same comment (if possible provide the figure) is to the chapter about the UPS-ROS interplay. Besides, explain TOC.
Response 9: Once again, we appreciate this good suggestion. Graphic illustration is always the best way to demonstrate molecular mechanisms in biology. We have added the figure (Figure 4) and explained TOC (Line 181)
Comment 10: I like Figure 3 and the pertinent text but then, there is a part of the text about plant HR (please check also if HR is explained) that is not sufficiently linked to the previous part
Response 10: Since both HR and SAR pathways are related to plant innate immune response, we feel it is logic to discuss these two pathways in the same section. We did explain HR together with SAR in the first paragraph of this section. The original sentence reads, “Instead, plants rely on a suite of sophisticated innate immune responses that generally result in two outcomes: systemic acquired resistance (SAR) and hypersensitive response (HR)” (Lines 475-477). Once again, taking this reviewer’s suggestions in Comments 8 and 9, we further provided a new figure (Figure 6) to illustrate the role of E3 ligases in HR pathway in the revision.
Comment 11: The text in the last chapter is difficult to follow. Partially because of the not precise language. For example, what the authors mean by ‘the number of UPS loci’? Besides, I do not understand why the main focus of the last chapter of this otherwise is a general review is on F-box proteins.
Response 11: We believe that the last chapter in our manuscript highlights the current challenges in the functional genomic studies of UPS genes and provides new insight into future directions. It is not our intent to make a general review on the F-box proteins. However, we hope that this reviewer can appreciate the large size of the F-box gene family in the plant UPS. It is our goal to share our evolution and genomics-based approaches in guiding functional genomic studies of active UPS genes. We apologize for the confusion of ‘the number of UPS loci’. It means “the number of UPS genes”. The same phrase has been used in the field, e.g. Mourad et al., J Exp Botany, 2006 (57): 3563–3573 and Vierstra, Nat Rev Mol Cell Biol 2009(10): 385-397. Nevertheless, to avoid this potential confusion, we have revised the sentence that now reads,”… converse to the large group of UPS genes discovered in plant genomes (Table 1; [5, 22]),…”(Lines 640-641). In addition, we added one sentence in the revision to explain our goal of this chapter. The new sentence reads, “Here, we use the F-box gene family as an example to illustrate future directions in tackling the role of the large group of UPS genes in plant growth and development, particularly in stress response” (Line 656-658). We hope that this reviewer finds this chapter exciting and important.
Reviewer 2 Report
This is a nice and comprehensive review article. I do not have any major criticism. If the author could include several sentences addressing the number of F-box proteins in other organisms such as yeast, nematode, fly, mice, and human, the readers may recognize again the diversity of plant UPS.
Author Response
Response to Reviewer2’s Comments
Comment 1: This is a nice and comprehensive review article. I do not have any major criticism. If the author could include several sentences addressing the number of F-box proteins in other organisms such as yeast, nematode, fly, mice, and human, the readers may recognize again the diversity of plant UPS.
Response 1: We appreciate the positive comments from this reviewer about our work in this review article. Based on the suggestion, we have provided the number of F-box genes predicted in yeast, nematodes, fly, and human (Lines 100-112).
Reviewer 3 Report
The manuscript “The Ubiquitin Switch in Plant Stress Response” is a review article explores the latest knowledge about protein turnover by the plant the ubiquitin (Ub)-26S proteasome system (UPS) and the genome evolution perspective of plant UPS and future directions. The manuscript is well-organized and suites the scope of the journal. I recommend doing these minor editing:
- Please align the first column in tables 1 and 2 to the left.
- I suggest using a different color than the purple for substrate in Fig 1.
Author Response
Response to Reviewer3’s Comments
Comment 1: The manuscript “The Ubiquitin Switch in Plant Stress Response” is a review article explores the latest knowledge about protein turnover by the plant the ubiquitin (Ub)-26S proteasome system (UPS) and the genome evolution perspective of plant UPS and future directions. The manuscript is well-organized and suites the scope of the journal. I recommend doing these minor editing:
Response 1: We appreciate the positive comments received from this reviewer about our work. Based on the suggestion, we have revised the manuscript accordingly.
Comment 2: Please align the first column in tables 1 and 2 to the left.
Response 2: Thanks for the comment. We have revised the format of the tables accordingly
Comment 3: I suggest using a different color than the purple for substrate in Fig 1.
Response 3: Thanks for this suggestion. We have changed the color to gray to distinguish substrate from other proteins illustrated in Figure 1.
Round 2
Reviewer 1 Report
Good job. The work is significantly better.